# Next-generation sequencing-based gene panel tests for the detection of rare variants and hypomorphic alleles associated with primary open-angle glaucoma

Elena Milla[1,2‡]*, Javier Laguna[3‡], Mª. Socorro Alforja[1], Beatriz Pascual[4], María José Gamundi[4], Emma Borràs[4], Imma Hernán[4], María Jesús Muniesa[1], Marta Pazos[1], Susana Duch[2], Miguel Carballo[4], Meritxell Jodar[3,5], on behalf of the EMEIGG group[¶]

1 Glaucoma Unit, Department of Ophthalmology, ICOF, Hospital Clínic de Barcelona, Barcelona, Spain,
2 Innova Ocular-ICO, Barcelona, Spain, 3 Department of Biochemistry and Molecular Genetics, CDB, Hospital Clínic de Barcelona, Barcelona, Spain, 4 Molecular Genetics Unit, Hospital de Terrassa, Barcelona, Spain, 5 Department of Biomedicine, Faculty of Medicine and Biomedical Sciences, University of Barcelona, Barcelona, Spain

‡ EM and JL contributed equally to this work and considered as first authors.
¶ Spanish Multicentre Glaucoma Group (Estudio Multicéntrico Español de Investigación Genética del Glaucoma, EMEIGG), Spain. Complete membership of the author group can be found in the Acknowledgments.
* millagrinoelena@gmail.com

**Data Availability Statement:** The minimal dataset is contained within the paper and its Supporting Information file.

## Abstract

Primary open-angle glaucoma (POAG) is a complex disease with a strong hereditably component. Several genetic variants have recently been associated with POAG, partially due to technological improvements such as next-generation sequencing (NGS). The aim of this study was to genetically analyze patients with POAG to determine the contribution of rare variants and hypomorphic alleles associated with glaucoma as a future method of diagnosis and early treatment. Seventy-two genes potentially associated with adult glaucoma were studied in 61 patients with POAG. Additionally, we sequenced the coding sequence of *CYP1B1* gene in 13 independent patients to deep analyze the potential association of hypomorphic *CYP1B1* alleles in the pathogenesis of POAG. We detected nine rare variants in 16% of POAG patients studied by NGS. Those rare variants are located in *CYP1B1*, *SIX6*, *CARD10*, *MFN1*, *OPTC*, *OPTN*, and *WDR36* glaucoma-related genes. Hypomorphic variants in *CYP1B1* and *SIX6* genes have been identified in 8% of the total POAG patient assessed. Our findings suggest that NGS could be a valuable tool to clarify the impact of genetic component on adult glaucoma. However, in order to demonstrate the contribution of these rare variants and hypomorphic alleles to glaucoma, segregation and functional studies would be necessary. The identification of new variants and hypomorphic alleles in glaucoma patients will help to configure the genetic identity of these patients, in order to make an early and precise molecular diagnosis.

**Funding:** The author(s) received no specific funding for this work.

**Competing interests:** The authors have declared that no competing interests exist.

## Introduction

It is estimated that almost 112 million people worldwide will have glaucoma in 2040, which will continue to be the second leading cause of blindness worldwide [1]. It is known that half of all glaucoma cases go undiagnosed, and therefore efficient screening methods for detecting glaucoma are essential. Genetic studies have enormously progressed in recent years with the advent of the Human Genome Project [2]. Increasingly, molecular genetics is seen as a diagnostic tool in daily clinical practice. Molecular genetic analysis could be used to detect the disease in its early stage (pre-symptomatic relatives), to perform phenotypic-genotypic correlations, to personalize treatment, to refine the prognosis of the disease and to offer genetic counselling [3].

Current next-generation sequencing (NGS) allows millions of DNA sequences to be produced in a single reaction, avoiding the limitations of the Sanger method of gene-to-gene and exon-to-exon sequencing [4]. In this way, through NGS, a large amount of data is obtained and a large number of variants will be observed in each individual. However, the classification of the gene variants is the challenge posed in current genetic molecular diagnosis.

Through linkage analysis, 23 loci and four genes (*MYOC/TIGR*, *CYP1B1*, *OPTN*, and *WDR36*) had already been associated with glaucoma in the past years. Early-onset glaucoma (<40 years) is more likely to be inherited according to a classic Mendelian pattern involving single genes, mainly *CYP1B1* in primary congenital glaucoma (PCG) and *MYOC* in juvenile open-angle glaucoma (JOAG), whereas glaucoma in adults tends to be more complex due to its multifactorial inheritance [5, 6]. Family grouping is a known risk factor for glaucoma in the adult, with a risk 1–10 times greater than the observed in general population among the first-degree relatives of an affected individual [7, 8]. Therefore, the surveillance of these individuals is indicated for early detection and treatment [9, 10].

However, these disease-causing genes mentioned above account for <10% of primary open angle glaucoma (POAG) cases in the general population [11]. Therefore, in the past decade, efforts have been made to elucidate the genetic causes of adult glaucoma. The application of advanced genetic technology has increased the list of candidate genes [6]. Recent Big Data studies using genome-wide association study (GWAS) have identified over 100 loci related with oxidative stress, DNA repair mechanisms, mitochondrial DNA genes, sex hormones, and phenotypic traits associated with glaucoma [12]. Quantitative endophenotypic agents that act as risk factors (pachymetry, axial length, anterior chamber depth. . .) and glaucomatous risk stratification have been identified [13, 14]. Therefore, genetic screening appears increasingly promising not only for PCG and JOAG diagnosis, but also for POAG.

In 2006, our research group created the Spanish Multicentre Glaucoma Group (*Estudio Multicéntrico Español de Investigación Genética del Glaucoma*, EMEIGG), including 18 Eye Hospitals throughout Spain. We performed an initial study to identify pathogenic variants in the *MYOC* and *CYP1B1* genes, which were the only fully-described glaucoma-causing genes at that time [15]. The aim of this study is to test a custom gene panel that gathered recently described candidate genes associated with POAG using NGS and to evaluate the impact of the implementation of NGS as a future method for diagnosis and early treatment of adult glaucoma. And, additionally, based on the NGS results, we set out to evaluate the impact of hypomorphic alleles of *CYP1B1* in a new cohort of patients with POAG.

## Materials and methods

### Patients

A total of 74 patients signed written informed consent to participate in this study and were asked to fill in a questionnaire including personal, biographic, demographic, family, and

clinical data. This study was carried out in accordance with the Declaration of Helsinki and approved by the Research Ethics Committee of the Hospital Clínic of Barcelona.

All participating ophthalmologists from the EMEIGG completed a standardized questionnaire to homogenize the data collection. A full history was taken, including systemic and ophthalmologic disease, family members affected by glaucoma (number and type of relative) and family history of consanguinity. POAG was diagnosed in the presence of compatible perimetric lesions correlated with typical glaucomatous changes of the optic nerve in patients over 40 years and absence of a cause for secondary glaucoma diagnosis. Patients affected with low-tension glaucoma (LTG) which is a form of POAG in which there is a glaucomatous optic neuropathy in the presence of intraocular pressures (IOP) lower than 20 mmHg were also included in the study. Patients with early-onset glaucomas who were diagnosed with PCG or JOAG were excluded.

Each referring ophthalmologist graded the disease stage, based on the ophthalmologic exams, the aspect of the optic disc, and visual field-testing results, and classified each case as initial, moderate, or severe glaucoma according to the Hodapp-Parrish-Anderson grading scale, which refers to visual field mean defect (MD) (initial if MD < −6 dB, moderate if MD between −6 and −12 dB, and advanced if MD > −12 dB). Data about the number of eye surgeries and current ocular hypotensive medication were recorded. The presence of thin pachymetry was also annotated as being an endophenotypic trait related to glaucoma conversion and severity.

## Study of glaucoma-related genes using NGS

Sixty-one patients (all unrelated, except two siblings) with POAG (or LTG) (Table 1) were included in the NGS study, with a mean age at diagnosis of 51 ± 12 years (range: 18–76 years). Although the study included adult glaucoma cases, four patients were younger than 40 years at the time of diagnosis (18–27 years), but did not display the typical features of JOAG, and were considered as early-onset POAG.

Thirty-nine genes potentially associated with POAG were studied (Table 2). After a thorough search in OMIM, Orphanet and Human Gene Mutation Database (HGMD), we selected

**Table 1. Clinical data of patients studied by NGS and Sanger sequencing.**

| | | | Patients NGS N = 61 | Patients Sanger N = 13 |
|---|---|---|---|---|
| **Gender** | **Female** | | 33 | 8 |
| | **Male** | | 28 | 5 |
| **Diagnosis** | **POAG** | **Initial** | 9 | 2 |
| | | **Moderate** | 14 | 8 |
| | | **Severe** | 31 | 3 |
| | **LTG** | **Severe** | 3 | - |
| | **Early-onset POAG** | **Moderate** | 1 | - |
| | | **Severe** | 3 | - |
| **Family history** | **Yes** | | 50 | 11 |
| | **No** | | 11 | 2 |
| **Surgery** | **Yes** | | 38 | 6 |
| | **No** | | 23 | 7 |
| **Ocular hypotensive medication** | **Yes** | | 15 | 1 |
| | **No** | | 46 | 12 |

LTG: Low-Tension Glaucoma; POAG: Primary Open-Angle Glaucoma.

**Table 2. List of genes included in the custom glaucoma panel designed.**

| ADRB1 | ERCC2 | OCLM |
|---|---|---|
| ADRB2 | GALC | OLFM2 |
| AGTR2 | GAS7 | OPA1 |
| ATOH7 | GSTM1 | OPTC |
| BCAS3 | HK2 | OPTN |
| BMP4 | LMX1B | PAX6 |
| CARD10 | MFN1 | PON1 |
| CAV1 | MFN2 | SIX1 |
| CAV2 | MTHFR | SIX6 |
| CDC7 | MYOC | TGFBR3 |
| CDKN2B | NCK2 | TMCO1 |
| COL8A2 | NOS3 | WDR36 |
| CYP1B1 | NTF4 | XRCC1 |

the classic known glaucoma-causing genes, genes associated with elevated IOP or POAG risk, anterior segment dysgenesis, glaucoma, glaucoma-related endophenotypic traits, optic nerve pathology or augmented susceptibility and retinal vessels anomalies and, finally, syndromic glaucoma-causing genes. Besides, additional genes have been included because a potential association with glaucoma has been suggested in some scientific literature studies [16–25].

Libraries were generated through gene capture by hybridization with the SeqCap EZ system (NimbleGene, custom panel "cl1_GM1") and subsequent massive parallel sequencing was carried out with a HiSeq™ 2000 platform sequencer (Illumina) at Sistemas Genómicos, S.L. Sequence reads were aligned with the reference genome GRCh38/hg38. Alignment was made using the BWA (Burrows-Wheeler Aligner) tool and scripts designed by Sistemas Genómicos, S.L. Genetic variants in coding and flanking sequences of genes with frequencies <1% were annotated. Annotated sequencing data was investigated using the filtering system for nonsense, missense, synonym and intronic variants located up to +10 bp in the flanking sequences. All exons and intronic regions up to +10 bp from the exons showed coverages greater than 20x. UTR regions were not analyzed. SNV annotation was made using ANNOVAR [26].

## Sanger sequencing of CYP1B1 in a new cohort of POAG patients

After observing the results of the NGS study, we included 13 additional patients for the study of the *CYP1B1* gene using Sanger sequencing. These 13 patients were not studied by NGS. The coding sequence of the *CYP1B1* gene was directly sequenced, following the same selection criteria as before (Table 1). The mean age at diagnosis was 55 ± 10 years (range: 40–79 years).

Specific oligonucleotides were designed to amplify the two coding exons of *CYP1B1* (NM_000104.4; S1 Table).

## Interpretation and classification of variants

*In silico* prediction algorithms included in Varsome (https://varsome.com/) were used to classify variants. The *in silico* tools used were: CADD, Polyphen2, DEOGEN2, MutPred, FATHMM-XF, Mutation assessor, MVP, PROVEAN, EIGEN, LRT, SIFT, BLOSUM, DANN, LIST-S2, M-CAP, MutationTaster and PrimateAI. ClinVar database was also consulted to check the classification reported by other subscribers.

General population frequencies and the highest frequencies in a population were obtained from gnomAD v3.1.2. Final variant interpretation was performed according to the American

College of Medical Genetics and Genomics (ACMG) and the Association for Molecular Pathology (AMP) recommendations [27]. In this way, we considered rare variants classified/ described as pathogenic, likely pathogenic or variant of uncertain significance (VUS) with low frequencies in the general population (<1%), as well as variants that have been described as hypomorphic variants.

## Results

### Rare genetic variants detected in POAG patients by NGS

We detected nine rare or hypomorphic variants (Table 3) in 10 patients (16%) by NGS (Table 4). These variants were found in seven of the glaucoma-related genes. All variants were detected in heterozygosity and classified as VUS. The allelic frequency of these variants in general population was very low (<1%) (Table 3). Most patients described with variants showed a severe POAG phenotype (70%, Table 4). Additionally, all patients with a rare genetic variant had a family history of glaucoma and most of them had required surgery or maximal ocular medication (Table 4). The presence of thin pachymetry readings was detected in two patients (patient 1 and patient 8).

The only variant detected in two unrelated patients was the variant p.Y81N in the *CYP1B1* gene (Table 4). Although different *in silico* algorithms and functional studies predict a possible pathogenicity of this variant, it has been identified in 528 individuals from genomes and exomes available in gnomAD, even in three of them in a homozygosis state. Nevertheless, the comparison of residues between organisms showed marked conservation of Y81 in different organisms (Fig 1) and its association with POAG has been reported several times [15, 28–31], being described as a hypomorphic variant in some cases [32–35]. Therefore, this variant was classified as a VUS.

Although familial studies of VUS are not recommended in clinical practice [36], the segregation study of the p.Y81N variant in the *CYP1B1* gene was performed in both families for

**Table 3. Description and classification of variants detected in this study.** All variants were detected in heterozygosity.

| Gene | Transcript | Variant (cDNA) | Variant (protein) | ACMG criteria | ClinVar classification (number of times) | ACMG classification | AF in general population | Highest AF and population |
|---|---|---|---|---|---|---|---|---|
| *CYP1B1* | NM_000104.4 | c.241T>A | p.Y81N | PS3, PP3, BS1, BS2 | VUS (2); LB (1); B (2) | VUS (Hypomorphic) | 0.35% | 1.36% European (Finnish) |
| *CARD10* | NM_014550.4 | c.1307C>T | p.T436M | PM2, BP4 | Not reported | VUS | 0.01% | 0.03% Ashkenazi Jewish |
| *CARD10* | NM_014550.4 | c.2952C>A | p.C984X | PM2, BP4 | Not reported | VUS | Not found | Not found |
| *MFN1* | NM_033540.3 | c.1601G>A | p.R534Q | PM2, PP3 | Not reported | VUS | Not found | Not found |
| *MFN1* | NM_033540.3 | c.2101G>T | p.E701X | PM2 | Not reported | VUS | Not found | Not found |
| *OPTC* | NM_014359.4 | c.893G>A | p.R298H | PM2 | Not reported | VUS | 0.03% | 0.21% Latino/ Admixed American |
| *OPTN* | NM_001008212.2 | c.1552C>T | p.Q518X | PM2 | Not reported | VUS | Not found | Not found |
| *SIX6* | NM_007374.3 | c.635C>T | p.T212M | PM2, PP3 | VUS (2) | VUS (Hypomorphic) | 0.02% | 0.03% European (non-Finnish) |
| *WDR36* | NM_139281.3 | c.892G>A | p.E298N | PM2, PP3 | Not reported | VUS | 0.01% | 0.11% Ashkenazi Jewish |
| *CYP1B1* ¶ | NM_000104.4 | c.83C>G | p.S28W | PM2, PP5, PS1 | Not reported | VUS (Hypomorphic) | 0.004% | 0.12% Ashkenazi Jewish |

¶Variant detected only in the study by Sanger sequencing. ACMG: American College of Medical Genetics; AF: allelic frequency; B: benign; LB: likely benign; NGS: next-generation sequencing; VUS: variant of uncertain significance.

**Table 4. Clinical characteristics of patients and variants detected in this study.** All variants were detected in heterozygosity.

| Patient | Gender | Age at diagnosis | Diagnosis | Features | Family history of glaucoma | Consanguinity | Surgery | Ocular hypotensive medication | Gene | Transcript | Variant (protein) |
|---|---|---|---|---|---|---|---|---|---|---|---|
| Next-generation sequencing | | | | | | | | | | | |
| Patient 1 | Female | 55 | POAG | Initial glaucoma, with PDS and myopia | Yes (mother, sister) | No | No | No | CYP1B1 | NM_000104.4 | p.Y81N |
| Patient 2 | Male | 76 | POAG | Severe glaucoma | No | No | Yes | Yes | CYP1B1 | NM_000104.4 | p.Y81N |
| Patient 3 | Female | 55 | POAG | Moderate glaucoma | Yes (daughter) | No | Yes | No | SIX6 | NM_007374.3 | p.T212M |
| Patient 4¶ | Female | 42 | POAG | Severe glaucoma, with retinal venous and arterial occlusions | Yes (brother) | No | Yes | No | CARD10 | NM_014550.4 | p.T436M |
| Patient 5¶ | Male | 30 | POAG | Severe glaucoma, with retinal venous and arterial occlusions | Yes (sister) | No | Yes | No | CARD10 | NM_014550.4 | p.T436M |
| Patient 6 | Female | 65 | POAG | Severe glaucoma | Yes (son, 2 siblings) | Yes | No | Yes | CARD10 | NM_014550.4 | p.C984X |
| | | | | | | | | | MFN1 | NM_033540.3 | p.R534Q |
| Patient 7 | Female | 65 | POAG | Moderate glaucoma | Yes (4 siblings, nephew) | No | No | Yes | OPTC | NM_014359.4 | p.R298H |
| Patient 8 | Male | 25 | Early-onset POAG | Severe glaucoma | Yes (father, brother) | No | Yes | No | OPTN | NM_001008212.2 | p.Q518X |
| Patient 9 | Male | 60 | LTG | Severe glaucoma | Yes (daughter) | No | Yes | No | WDR36 | NM_139281.3 | p.E298N |
| Patient 10 | Female | 68 | POAG | Severe glaucoma | Yes (father, brother) | No | Yes | No | MFN1 | NM_033540.3 | p.E701X |
| Sanger sequencing of CYP1B1 | | | | | | | | | | | |
| Patient 11 | Male | 58 | POAG | Moderate glaucoma | No | No | Yes | No | CYP1B1 | NM_000104.4 | p.Y81N |
| Patient 12 | Female | 60 | POAG | Initial glaucoma | Yes (maternal grandmother) | No | No | No | CYP1B1 | NM_000104.4 | p.Y81N |
| Patient 13 | Male | 60 | POAG | Moderate glaucoma and myopia | Yes (mother, 2 siblings) | No | No | No | CYP1B1 | NM_000104.4 | p.S28W |

¶Siblings. LTG: low-tension glaucoma; NGS: next-generation sequencing; PDS: pigment dispersion syndrome; POAG: primary open-angle glaucoma.

research purposes (Fig 2). The variant segregated with the disease in the family of patient 1. The patient's sister, who was also diagnosed with POAG, carried the p.Y81N variant in *CYP1B1* in heterozygosity. The variant was also detected in the patient's daughter, who had not signs of glaucoma, but has ocular hypertension. The family of patient 2 was smaller and not informative. The segregation study showed that the patient's daughter, who had no signs of glaucoma or ocular hypertension, did not have the variant. His wife also had glaucoma, but she also underwent NGS testing, and no variant was detected.

We also observed two siblings with the variant p.T436M in the *CARD10* gene. Both siblings (a 45-year-old female and a 43-year-old male) had a very aggressive POAG phenotype (Fig 3),

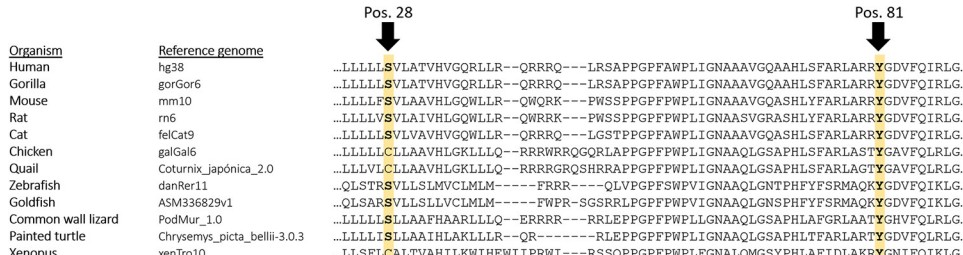

**Fig 1.** Comparison of the evolutionary conservation of p.S28W (left) and p.Y81N (right) in the *CYP1B1* gene in the reference genome of some organisms.

with very high IOP and pigment dispersion syndrome (pigment dots scattered around the corneal endothelium and pigmented trabecular meshword), although other characteristic features of classic pigmentary glaucoma (typical Krukenberg spindle and iris atrophy) were missing. Both also developed retinal venous and arterial occlusions. Aside from the siblings, we detected another variant in the *CARD10* gene (p.C984X) in a 65-year-old female, along with a second variant in the *MFN1* gene (p.R534Q). This patient had a severe POAG phenotype, and was the only patient in the study with a family history of consanguinity (grandparents were first degree cousins). Regarding the *MFN1* gene, we also detected another variant (p.E701X) in a 68-year-old patient with severe glaucoma. Both variants in the *CARD10* gene and in the *MFN1* gene have not been reported in ClinVar and most of them do not appear in gnomAD, except for the variant p.T436M in the *CARD10* gene that was identified in 12 individuals in heterozygosity. All of them have one or two deleterious predictors and should therefore be classified as VUS.

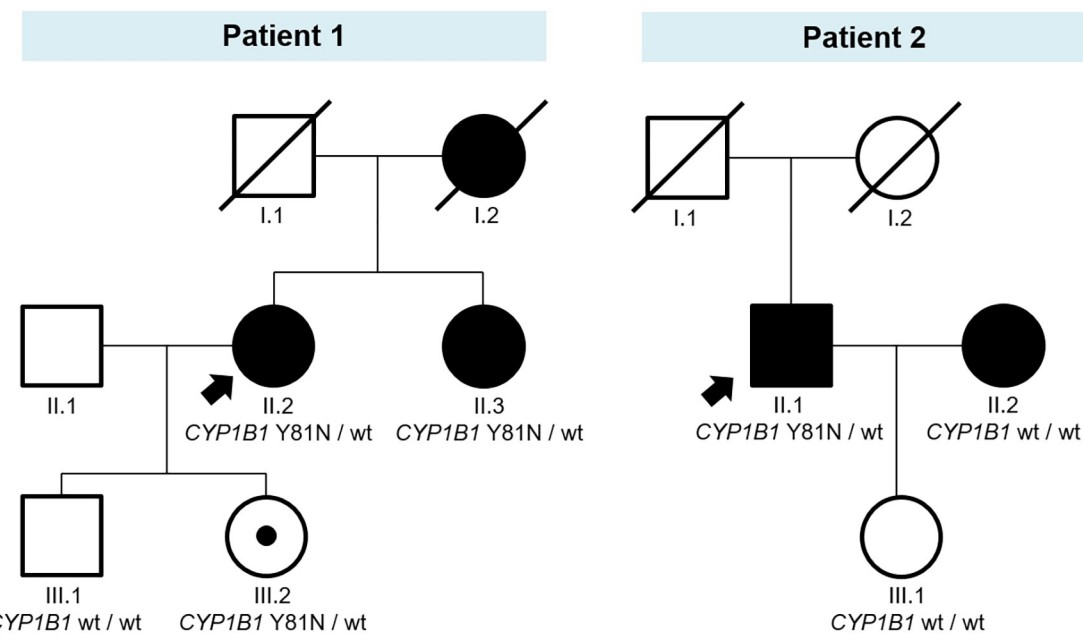

**Fig 2.** Pedigrees of patients in whom the p.Y81N variant in the *CYP1B1* gene was detected in the NGS study: patient 1 (left) and patient 2 (right). Arrows indicate the index case. Black symbols indicate glaucoma phenotypes, and carriers of the variant are indicated by black dots in the symbols. wt: wild-type allele.

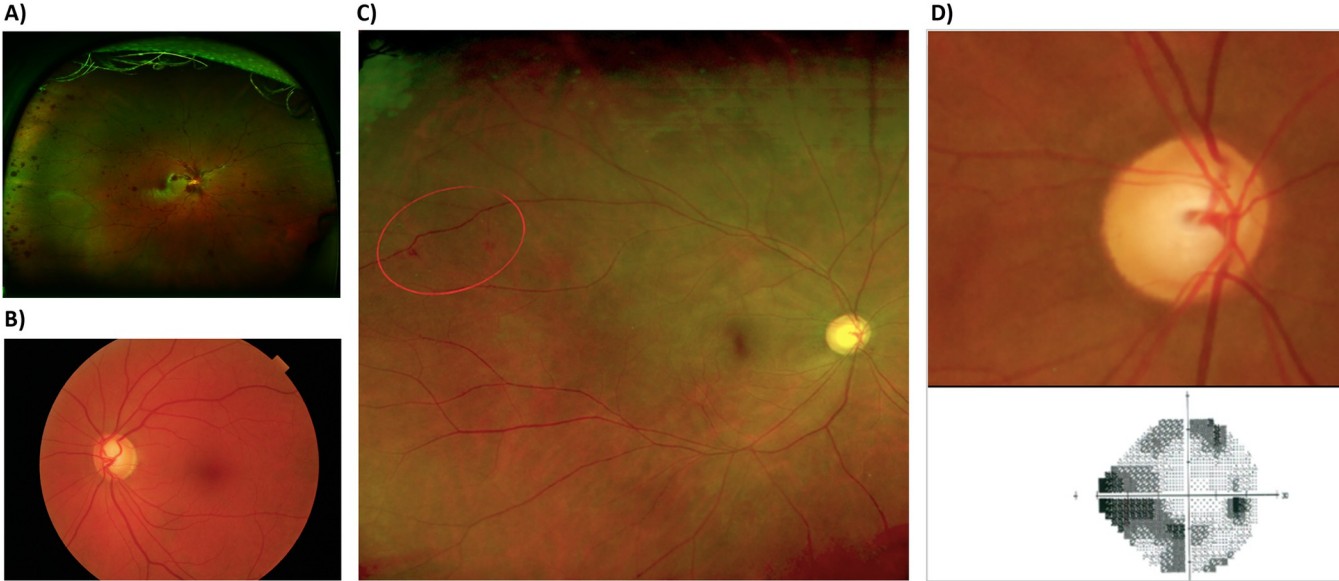

**Fig 3. Ophthalmologic studies in patients 4 and 5 (siblings) with the variant p.T436M in the *CARD10* gene. A**: Optomap image of ocular fundus of the patient 4 showing macular whitening and a myriad of splinter retinal hemorrhages and venous tortuosity secondary to central retinal vein occlusion with cilioretinal artery occlusion in the right eye. **B**: Funduscopic image of the left eye of the patient 4 showing severe optic disk cupping in the context of advanced glaucoma with pigment dispersion syndrome. **C**: Optomap image of the right eye fundus of the patient 5 showing mild macular whitening with scant peripheral hemorrhages one month after non-ischemic central retinal vein occlusion associated with cilioretinal artery occlusion in the right eye. **D**: Retinography of the patient 5 (up) showing severe right optic disk cupping causing advanced visual field constriction seen on perimetry (below).

The other genetic variants were identified in *OPTC* (p.p.R298H), *OPTN* (p.Q518X), *SIX6* (p.T212M), and *WDR36* (p.E298N). As shown in Table 3, all these variants have very low allele frequencies in general population and most of them have not been described in ClinVar. It is interesting to note that one of these variants, p.T212M in the *SIX6* gene, has also been reported in the literature as hypomorphic variant [37].

## Analysis of hypomorphic *CYP1B1* variants

It has been reported that *CYP1B1* may play a role in the pathogenesis of POAG in a significant proportion of cases [30, 31, 34]. For this purpose, we sequenced the coding region of this gene in a new cohort of POAG patients. Heterozygous missense variants were detected in three POAG patients (23%, Table 4). None of these patients had severe glaucoma. The aforementioned variant p.Y81N in the *CYP1B1* gene was also detected in two patients of these three patients (Table 4).

The variant c.83C>G (p.S28W) in the *CYP1B1* gene was identified in a 66-year-old male patient, diagnosed with myopia magna and moderate glaucoma at the age of 60 (Fig 4). His mother and two siblings were also affected by glaucoma. He was taking ocular hypotensive medication, but his IOP remained very high despite maximal treatment. Finally, surgery was necessary in both eyes. To the best of our knowledge, this variant has never been reported in ClinVar and has been detected at a very low frequency in the general population (0.004%). According to the ACMG/AMP recommendations, the variant was classified as VUS following criteria: PM2 (absent from controls in Exome Sequencing Project, 1000 Genomes Project, or Exome Aggregation Consortium) and PP5 (reputable source recently reports variant as pathogenic, but not functional experimental evidence). Similarly to the variant p.Y81N, the comparison of residues between different organisms showed a marked conservation of S28 in mammals and other organisms (Fig 1).

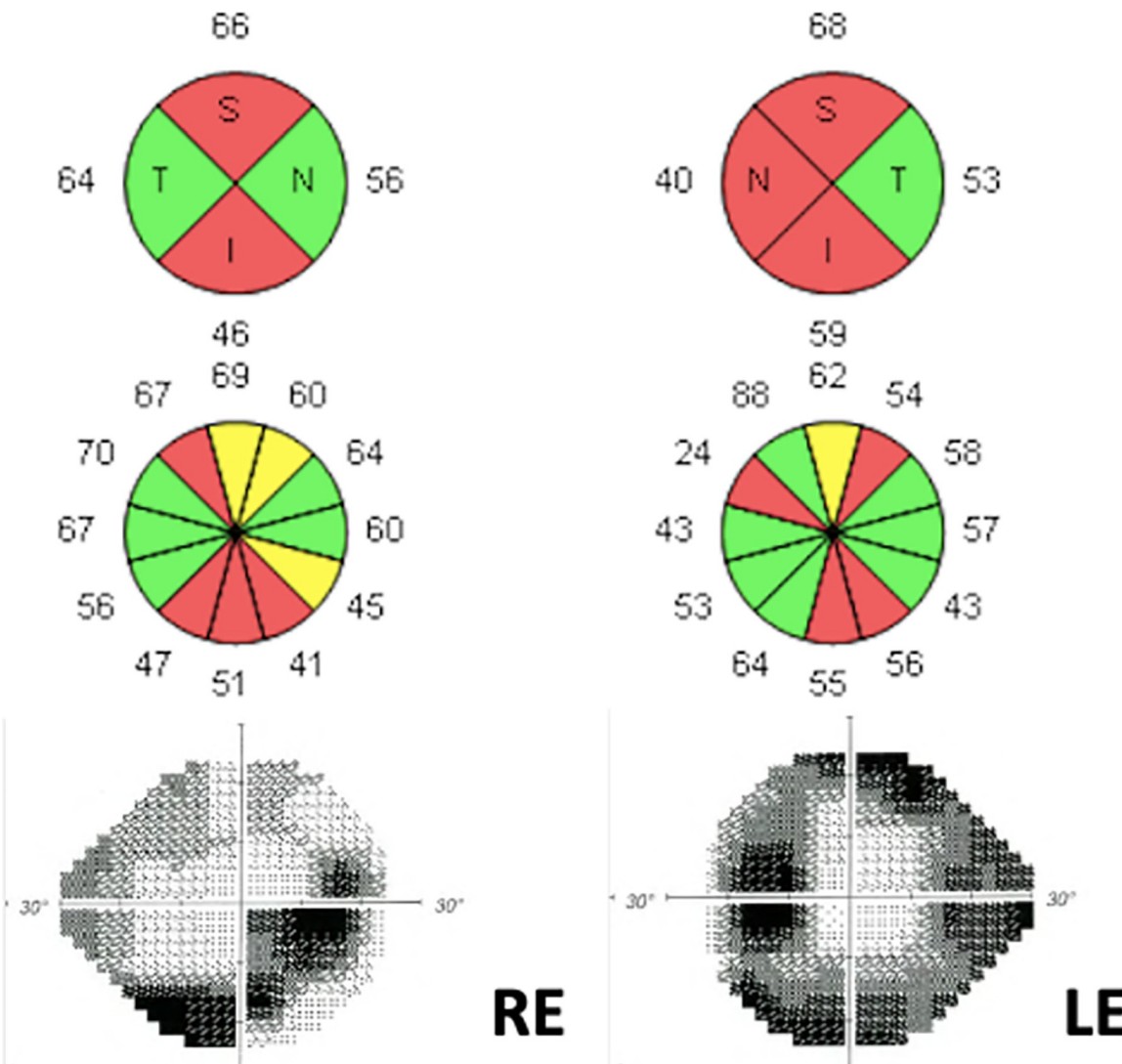

**Fig 4. Ancillary tests of patient 13.** This figure shows the structural and functional damage of the right (RE) and left eye (LE). The optical coherence tomography shows severe peripapillary retinal nerve fiber layer thinning that correlates with the visual field findings that depicts a marked inferior arcuate scotoma for the RE and tubular island of vision for the LE.

## Discussion

According to Lander and Schork [38], the genetic study of adult glaucoma is complex since its transmission mechanisms are sometimes unclear and have variable penetrance and late onset. Adult glaucoma is a multifactorial genetic disease whose outcome is also influenced by a number of environmental factors, many of them unknown. In addition, there are several ethnic and geographic disparities between populations, which further complicate the genetic study of glaucoma [39]. Thanks to recent technological improvements, such as NGS, the list of candidate genes possibly associated with glaucoma has increased considerably since the initial Sanger direct sequencing studies, when only four genes showed a strong association with glaucoma [6].

In this study, we applied NGS to study a cohort of patients with POAG. Hypomorphic alleles and rare variants that were not previously described in the literature or with a very low

population frequency have been detected in 10 out of 65 patients assessed. The variant p.Y81N in *CYP1B1* gene was the most frequent variant without considering related patients by NGS (two patients). Given the intimate relationship between *CYP1B1* and glaucoma and the results observed in the first group, we sequenced the *CYP1B1* gene in a different cohort of patients, detecting the variant p.Y81N in two more POAG patients. Thus, the variant p.Y81N was detected in four unrelated patients (two patients from the NGS study and two patients from the Sanger study).

CYP1B1 is an enzyme involved in drug, fatty acid and steroid metabolism located in the endoplasmic reticulum membrane, peripheral membrane protein and microsome membrane. CYP1B1 participates in the metabolism of an as-yet-unknown biologically-active molecule that participates in eye development. Genetic variants in this gene had always been associated with PCG, an autosomal recessive inherited trait [40, 41]. However, in recent years it has been observed that CYP1B1 also could play an important role in the development of adult glaucoma [30, 31]. Hypomorphic alleles pose a challenge in the interpretation of genomic variants [42]. A hypomorphic allele results in a partial loss of the normal (wild-type) gene function, often characterized by reduced expression of the gene product (protein or RNA), although this reduction does not reach a 100% reduction in normal gene function. In the literature we found few articles about hypomorphic alleles in POAG, and most of them focus on genetic variants in *CYP1B1* [31–33, 43, 44] and *SIX6* [37]. In the current study, hypomorphic variants in *CYP1B1* and *SIX6* genes have been identified in 8% of the total POAG patient assessed. As commented before, the genetic variant p.Y81N has been identified in heterozygous state in four unrelated patients. This genetic variant has been reported at a low frequency (0.35%) in the general population and has been described as a heterozygous hypomorphic variant for POAG [32–35], with a significantly reduced enzymatic activity and reduced protein stability (18–40% of the wild-type activity) [32]. In one of the families, the segregation study revealed the presence of a variant in an individual who had not yet shown signs of glaucoma, but who had ocular hypertension. These results suggest that it might be interesting to propose familial studies in patients with *CYP1B1* hypomorphic alleles to identify asymptomatic carriers of the variant for close ophthalmological follow-up.Besides, we also identified for the first time the variant c.83C>G (p.S28W) in the *CYP1B1* gene in a patient with a moderate glaucoma and myopia. Interestingly, a different genetic variant but at the same position (c.83C>T) which produces the same amino acid change as the variant found in our study (p.S28W) has been described as a hypomorphic variant and reported in patients with POAG [45] and PCG [46].

Both variants identified in our study in the *CYP1B1* gene caused amino acid changes and affected conserved residues located in structural domains, having the potential to modify enzyme activity by incorrect insertion of the protein into the endoplasmic reticulum membrane (p.S28W) and modification of substrate binding (p.Y81N) [45].

Additionally, we detected the variant p.T212M in the *SIX6* gene in a patient with a moderate glaucoma. This variant was also described as a hypomorphic variant. SIX6 is a transcription factor with a known function in the retinal progenitor cell proliferation during the eye development [47]. *In vivo* analyses in zebrafish have been useful to study the effect of this variant in the eye. Modified animal models with this variant have been found to have a smaller eye size, hypothesizing that the presence of hypomorphic alleles in *SIX6* could reduce the number of retinal ganglion cells and increase the risk of glaucoma [37].

Regarding the rare variants detected in other genes, we have observed variants in *CARD10*, *MFN1*, *OPTC*, *OPTN* and *WDR36*. Most of them have not yet been associated with glaucoma. Additional studies, including functional and segregation analyses, are imperative to confirm whether these genes and variants are indeed associated with the disease. Interestingly, we found two siblings with the variant p.T436M in the *CARD10* gene. This variant has not been

reported in the general population, but *in silico* tools predict a benign impact. Nevertheless, the variant is still considered a VUS. *CARD10* encodes for a caspase recruitment domain-containing protein, which is a signaling protein in the regulation of the NF-κB (nuclear factor kappa B) pathway. Since NF-κB is involved in the regulation of cellular apoptosis, it is likely that there is a relationship between *CARD10* and cell apoptosis, especially retinal ganglion cell apoptosis, producing higher optic nerve susceptibility to IOP elevations and POAG [48, 49]. In fact, Zhou et al. [18] observed a higher frequency of variants in this gene (4.28%) in patients with severe POAG compared to the control population (0.27%).

Besides, we found one patient with severe glaucoma and two VUS in *CARD10* and *MFN1*. In these cases, the use of polygenic risk scores may be of interest. These scores have been reported in recent years as promising for stratifying individual risk and prognosis of POAG [50, 51], although further studies are still needed.

The association of POAG with some of the rare variants identified in this study should be treated with caution. For example, we detected the nonsense variant p.Q518X in the *OPTN* gene in one patient with POAG. Recent studies have reported that the missense variant p. E50K is the only known variant in the *OPTN* gene associated with glaucoma, whereas loss-of-function variants in this gene are associated with amyotrophic lateral sclerosis (ALS) [52]. However, there are cases where ALS and glaucoma coexist, as seen in one of the patients reported by Maruyama *et al.* [53]. Therefore, variants in the *OPTN* gene may contribute to some additional risk of glaucoma in certain patient populations.

Similarly, contradictory results are reported for the association of pathogenic variants in the *WDR36* gene with glaucoma. Whereas several studies have indicated that genetic variants in *WDR36* gene are contributing risk factors for glaucoma progression and severity [54–58], recent clinical trials and meta-analyses have suggested a lack of effect [59–61]. However, it is known that this gene has a connection to glaucoma susceptibility and even to retinal homeostasis [62, 63]. Monemi *et al.* demonstrated *WDR36* gene expression in the lens, iris, ciliary muscles, ciliary body, trabecular meshwork, retina, and optic nerve by RT-PCR with four pathogenic variants (p.N355S, p.A449T, p.R529Q and p.D658G) associated with adult-onset POAG with implications for both high- and low-pressure glaucoma [64]. In our study, we detected the variant p.E298N in the *WDR36* gene in a patient with severe LTG, whose phenotype was very similar to another reported LTG patient with a variant (p.N355S) in the *WDR36* gene [58].

Some of the patients carrying rare variants (patient 1 and patient 8) presented with thin pachymetry, which is a known endophenotypic trait that increases the degree of severity of the glaucoma cases, whether this trait has been directly linked to the variant described or is a consequence of other genetic/epigenetic influences still has to be elucidated. Further studies are warranted to explore this relationship more comprehensively.

Certainly, addressing the limitations of our study is crucial. While NGS is a powerful tool for detecting genetic variations, it may have limitations in detecting copy number variations (CNV), including deletions or duplications. We acknowledge that our study focused primarily on single nucleotide variations and small indels, and we did not specifically investigate CNV. This limitation is important to consider, especially in cases where CNV could be contributing to the disease [65]. Future research efforts could explore complementary techniques or arrays designed for CNV detection to provide a more comprehensive genetic assessment of glaucoma-related variations.

Our findings suggest that NGS could be a valuable tool for the genetic assessment of glaucoma. However, to irrefutably show the contribution of these rare variants and hypomorphic alleles to glaucoma, additional studies, including functional evidence, will be necessary. The progressive identification of new rare variants and hypomorphic alleles in patients clinically

diagnosed with glaucoma will help to configure the genetic identity of these patients, in order to make an early and precise molecular diagnosis. And as a clinical application of these findings, the presence of hypomorphic alleles in asymptomatic relatives of our glaucoma patients acts, in our opinion, as a red flag that suggests close monitoring of these patients and early treatment decision in case of glaucoma suspicion.

## Supporting information

**S1 Table. Primers designed to amplify coding region of the *CYP1B1* gene.**
(DOCX)

## Acknowledgments

We would like to thank the rest of members of the EMEIGG group (*Estudio Multicéntrico Español de Investigación Genética del Glaucoma*): Esperanza Gutiérrez and Marta Montero (Hospital Doce de Octubre, Madrid); Cristina Vendrell (Hospital de Viladecans, Barcelona); José Abreu (Hospital Universitario de Canarias); Carmen Cabarga (Hospital Ramón y Cajal, Madrid); Soledad Jiménez (Hospital Universitario Puerta del Mar, Cádiz); Miguel Ángel Almela (Hospital Lluis Alcanyís, Xàtiva); Jordi Loscos (Hospital Universitari Germans Trias i Pujol, Badalona); Carlos Martínez Bello (Hospital Dos de Maig, Barcelona); Sergio Torregrosa (Hospital Punta de Europa, Cádiz); Rosa Martínez (Hospital Infanta Leonor, Madrid); Tiburcio Ibáñez (Hospital San Agustín, Linares); Lourdes Iglesias (Hospital Universitario La Princesa, Madrid); Pere Viñallonga (Institut Oftalmològic, Menorca); Lluís Soler (Hospital de Manresa, Barcelona); and Carmen Carrasco (Hospital de Alcorcón, Madrid).

## Author Contributions

**Conceptualization:** Elena Milla, Meritxell Jodar.

**Data curation:** Elena Milla, Javier Laguna, Mª. Socorro Alforja, Beatriz Pascual, María José Gamundi, Emma Borràs, María Jesús Muniesa, Marta Pazos, Susana Duch.

**Formal analysis:** Elena Milla, Beatriz Pascual, María José Gamundi, Miguel Carballo.

**Investigation:** Elena Milla, Javier Laguna, Mª. Socorro Alforja, Meritxell Jodar.

**Methodology:** Elena Milla, Javier Laguna, Beatriz Pascual, Emma Borràs, Imma Hernán, Meritxell Jodar.

**Project administration:** Elena Milla, Susana Duch, Miguel Carballo.

**Supervision:** Elena Milla.

**Validation:** Elena Milla, Javier Laguna, Meritxell Jodar.

**Visualization:** Elena Milla.

**Writing – original draft:** Elena Milla, Javier Laguna, Miguel Carballo.

**Writing – review & editing:** Elena Milla, Meritxell Jodar.

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
