## [Decision Letter · Decision Letter 0]

19 Sep 2023

PONE-D-23-03532Next-generation sequencing-based gene panel tests for the detection of rare variants and hypomorphic alleles associated with primary open-angle glaucomaPLOS ONE

Dear Dr. MILLA,

Thank you for submitting your manuscript to PLOS ONE. After careful consideration, we feel that it has merit but does not fully meet PLOS ONE’s publication criteria as it currently stands. Therefore, we invite you to submit a revised version of the manuscript that addresses the points raised during the review process.

We look forward to receiving your revised manuscript.

Kind regards,

Alvaro Galli

Academic Editor

PLOS ONE

Journal Requirements:

a) Did participants provide their written or verbal informed consent to participate in this study?

4. One of the noted authors is a group or consortium: EMEIGG group

In addition to naming the author group, please list the individual authors and affiliations within this group in the acknowledgments section of your manuscript. Please also indicate clearly a lead author for this group along with a contact email address.

5. Please upload a copy of Supporting Information Table 1 which you refer to in your text on page 8.

Reviewers' comments:

Reviewer's Responses to Questions

**Comments to the Author**

1. Is the manuscript technically sound, and do the data support the conclusions?

Reviewer #1: No

Reviewer #2: No

Reviewer #3: Partly

Reviewer #4: Yes

2. Has the statistical analysis been performed appropriately and rigorously? 

Reviewer #1: No

Reviewer #2: N/A

Reviewer #3: Yes

Reviewer #4: N/A

3. Have the authors made all data underlying the findings in their manuscript fully available?

Reviewer #1: No

Reviewer #2: Yes

Reviewer #3: Yes

Reviewer #4: Yes

4. Is the manuscript presented in an intelligible fashion and written in standard English?

Reviewer #1: Yes

Reviewer #2: Yes

Reviewer #3: Yes

Reviewer #4: Yes

5. Review Comments to the Author

Reviewer #1: The approach taken is interesting and there is useful data here, but there is much work to do before this is of a standard required for publication in a peer reviewed journal. My criticisms fall into two areas.

Firstly a general comment. The paper as written confuses the search for potentially Mendelian or near Mendelian alleles (the latter would be what I think they mean when they say hypomorphic alleles) with a search for association. If this is considered a search for alleles of large effect size then it is valid, with the proviso, as they rightly say, that these results are tentative and would be given more credence if they were replicated in other screens, as well as confirmed by segregation in families or functional tests. However, the use of chi-squared tests suddenly make it feel like an association screen. I am no statistical geneticist, but even I know that those stats are meaningless. They have performed thousands of tests - arguably every nucleotide they sequenced is a test, or at least every variant from reference they examined. If you do a thousand tests then your p value needs to be below 0.05/1000 before it is statistically significant. So this distinction needs to be discussed and made crystal clear, and those chi-squared tests need to be removed.

Secondly there is much information missing or in need of clarification.

The cohort they study is not well described. The section in mat/mets only talks about the 65 patients, not the 34, so reading that I get the impression the 34 are a subset of the 65, but on lines 97 and 296 these are clearly referred to as a separate cohort. Also it seems bizarre that they would weaken a genetic study by making their cohort so diverse - surely there are plenty of POAG patients across 18 Spanish centres without introducing heterogeneity by including NTG and OHT? All patients are said to be unrelated - was any attempt made to recruit affected relatives where possible, as this would allow segregation studies? In the introduction the authors rightly suggest genetic effects may be enriched in patients with early onset or strong family history - was any attempt made to recruit particularly among those groups? Also, if there is no overlap between the 65 patients and the possibly additional 34, the differences between them deserve some note. The second group seem milder, with more OHT and less surgery.

The process by which candidate genes were selected for screening is also poorly described. The authors simply say they looked at Pubmed and chose these 72 genes. Why these and not others, what criteria were used to select candidates? Also when was the selection made, because Pubmed is a rapidly changing and developing resource that may well give a very different set of candidates if they look again now. There are the obvious ones in their list but there are a few for which I was unaware of a link. I did also wonder why a couple of big hits from GWAS were not included - TCMO1, FOXC1, GAS7 and the CDKs are in this list, but not AFAP1 or ABCA1? I'm not asking for a one by one justification, just some objective criteria.

There are also gaps in the technical data. How many nucleotides did exons of the 72 genes amount to, how much flanking DNA was included, were 5' and 3' sequences included in the sequencing, how much intronic, 5' and 3' sequence was analysed, what kind of HiSeq was used, what depth of coverage was obtained?

Finally there was very little said about what was found other than the variants cherry-picked for presentation. I think we can safely presume that the nine variants presented were not the only things they found. There needs to be a brief summary statement of everything identified, then a list of everything that passed certain minimal criteria that need to be clarified - below 1% in the population (clear), in the coding sequence and +/-10bp (not clear), non-synonymous (not clear) then presumably predicted VUS, likely pathogenic or pathogenic (again not clear) - should be attached in a supplementary file. Lastly they need some further explanation of why only these nine were chosen for further examination.

In addition I have a few mostly minor corrections.

On line 68 and below "Through linkage analysis, 23 loci and four genes.... (ref 5)"; this is clumsy, they seem to be saying at a certain time there were only these findings then later there were more, but the "later" they quote is reference 7 that was actually published in 2016, a year before reference 5?

Line 130 - they need to decide whether they prefer NTG or LTG and stick with one. As far as I know these are not two different diseases but different names for one.

Legend to table 3 needs to state that (I assume) all variants were heterozygous.

Were mutations found in NGS rechecked by Sanger? Perhaps not essential with very modern NGS machines but if the data were older it might be reassuring, and also allows segregation if other family members are available.

Figure 1 is described in the text as homology "modelling". To me modelling means something different, this is just comparing evolutionary conservation of these residues, and since it only compares across mammals it isn't very conclusive - are these residues conserved in birds, reptiles, amphibians or fish? If not it would be useful to know.

Figure 2d is too small and fuzzy to derive anything from it.

Line 284 - "Adult glaucoma is a multifactorial genetic disease upon which a series of environmental factors must hatch" - clumsy - I suggest "Adult glaucoma is a multifactorial genetic disease the outcome of which is also influenced by a series of environmental factors" - or similar.

Also in the discussion there is much talk of hypomorphic alleles - a clear definition of what the authors mean by that would be useful.

Reviewer #2: The authors analysed a cohort of 65 patients with primary open-angle glaucoma (POAG) using a custom gene panel of 72 genes associated with the disease, to test the use of NGS as a diagnostics method. They reported 9 variants in 7 genes in 15% of patients. However some of the genes included in the panel have been reviewed and showed to have no association to POAG and none of the reported variant presented evidence to support a pathogenic impact. Additionally, the cohort is quite small to identify variants that contribute to <10% of the disease, especially with no variants reported in the main contributor to POAG (e.g. MYOC). Therefore, the evidence presented in this study does not support NGS for genetic diagnosis of glaucoma and does not support the conclusions.

Although variants in Mendelian genes account for around 5% of POAG cases, this is mainly driven by MYOC. It is surprising that no variants were reported in MYOC in this study. Have the authors identified any variants in the gene in their cohort?

Some of the genes included in the panel such as WDR36 and NTF4 have been reclassified by initiatives conducting evidence-based reviews (e.g. ClinGen & PanelApp) as having no association with POAG. Similarly, some rare variants in genes such as EFEMP1 and TEK have been associated with JOAG/POAG in a monogenic manner but have not been included in the list. The list of genes associated with POAG needs to be revised for proper interpretation of results in the context of using NGS as a diagnostic strategy.

Specifically, for WDR36 the initial variants reported in the literature were found to be too common in the general population, a meta-analysis showed no association in case-controls, the gene is ubiquitously expressed therefore expression in eye tissues does not constitute supporting evidence and no functional evidence inc. animal models reported a glaucoma phenotype. However, none of this is presented in the discussion, which only highlights evidence for an association to POAG. This gene has now been reclassified as having no association and should not be included in a gene panel for POAG assessing NGS for diagnostics.

Most of the genes included in the panel have been associated with POAG or its endophenotypes in genome-wide association studies, however rare variants have not yet been associated with the disease for most genes. The rare variants identified in these genes need to be discussed in this context, highlighting that additional studies, including functional evidence, would be needed before these addition of these genes to gene panels is justified.

A definition of hypomorphic variants is missing. In general, hypomorphic alleles refer to variants that can be common, do not cause disease on their own but functional evidence supports an impact. It is unclear from the paper which of the variants identified are rare vs hypomorphic, with evidence to support classification.

Please specify what in silico tools were used for variant impact prediction.

In addition to the above comments, the limitations need to discuss the detection (or lack of) of CNVs using NGS and its impact on detecting deletions or duplications associated with glaucoma (e.g. TBK1 duplications are known to cause POAG).

Other comments:

Some references are missing in the text while others reference incorrect papers. line 55 reference 1 does not report on the rate of undiagnosed glaucoma. line 69 reference 5 does not report on the loci and genes associated with POAG. line 86 ref 10 reports on optic disc parameters but no references are provided for the endophenotypes listed (AL, ACD, CCT). line 303 ref 27 and 28 did not report the first association of CYP1B1 to PCG. There are no references provided for the prevalence of glaucoma (line 53), the familial risk (line 75), the gene contribution to POAG (line 78), the 100 loci reported for POAG (line 84).

Table 3:

- Please note that the use of the ACMG PP5 and BP6 criteria has been discontinued (PMID 29543229) and should be removed for CYP1B1 Y81N.

- PVS1 only applies when LoF is a known disease mechanism. Therefore it does not apply to OPTN variants which cause POAG through a gain of function mechanism since LoF variants such as nonsense are not expected to cause glaucoma. Therefore, there is no evidence to support a role of the nonsense OPTN variant reported with POAG. ClinGen recently reviewed the association of OPTN to POAG and concluded that the only variant showing an association is E50K (https://search.clinicalgenome.org/kb/gene-validity/CGGV:assertion_1d8edc80-413c-4c76-bf06-89198175ac65-2022-05-10T170000.000Z)

- All in silico tools (CADD, SIFT, PP2, REVEL, Mutation Taster) predict a benign impact for CARD10 T436M which should then meet BP4. Please revise and acknowledge in the discussion.

- Please report the highest AF in a population instead of the general population as some of these variants are more common in some populations than others, which should be used when classifying variants.

- Add the ACMG classification for each variant based on evaluation in Table 3.

- Move nucleotide nomenclature to Table 3 instead of Table 4.

- CYP1B1 S28W is missing from Table 3.

-

Table 4:

- OPTN R298H in Patient 8 should be Q518*.

The prevalence of glaucoma in 2040 reported by Tham et al. is 112 million, not 120 million.

line 127 the range of age at diagnosis is listed as 25-76y but line 129 reports a patient diagnosed at age 18y.

line 267 CYP1B1 S28W is in gnomAD, please correct.

Reviewer #3: The authors have screened two different cohorts of POAG patients for disease causative mutations by using two different techniques; NGS panel for 72 genes associated with POAG and Sanger sequencing for Cyp1b1, a frequently mutated gene in patients with glaucoma of different types. The main aim is the applicability of custom NGS panel for POAG. The study is well planned and methods are up to date. The results are as per expectations but the interpretations need some more work to reach a definite conclusion. Following are some points for improvement of the draft

1. For first cohort of 65 patients, subjected to NGS, 53 patients have family history. It is important to check segregation of the alleles in other family members, to confirm its association with the disease.

2. The most common gene was CYP1B1 as per results. The non-penetrance of various CYP1B1 alleles is reported and its diagnosis in a proband need segregation studies as well to establish its pathogenic role.

3. Why CARD10 gene was not selected for sanger sequencing as it is second common gene as per NGS results

4. Authors should discuss the findings in the light of already reported data of the glaucoma genes in Spanish patients

Reviewer #4: I would like to make several recommendations which the authors may find useful to improve their paper:

• As the authors pointed out several times the importance of genetic diagnosis of glaucoma, I would recommend classifying detected variants (e.g. in table 3, not only ClinVar data). Additionally, in the same table, the ACMG criteria are listed in the column “Predicted effect in silico”. It is not clear are those criteria taken from Varsome database or the authors selected those by themselves (which I would recommend).

• Lines 77-78. Missing references.

• Lines 191-192. What variants were found correlated?

• The variant c.83C>G (p.Ser28Trp) in the CYP1B1 gene was not presented in tables 3 and 4.

• Lines 255-260. It is misleading how many variants were detected in CYP1B1 (2 in the tables, here one more, 4 in discussion...)

• Lines 307-309. Four patients with variants in CYP1B1?

6. PLOS authors have the option to publish the peer review history of their article (what does this mean?). If published, this will include your full peer review and any attached files.

Reviewer #1: No

Reviewer #2: No

Reviewer #3: No

Reviewer #4: **Yes: **Milena Jankovic

---

## [Author Response · Author response to Decision Letter 0]

2 Nov 2023

Reviewer #1: The approach taken is interesting and there is useful data here, but there is much work to do before this is of a standard required for publication in a peer reviewed journal. My criticisms fall into two areas.

Firstly a general comment. The paper as written confuses the search for potentially Mendelian or near Mendelian alleles (the latter would be what I think they mean when they say hypomorphic alleles) with a search for association. If this is considered a search for alleles of large effect size then it is valid, with the proviso, as they rightly say, that these results are tentative and would be given more credence if they were replicated in other screens, as well as confirmed by segregation in families or functional tests. However, the use of chi-squared tests suddenly make it feel like an association screen. I am no statistical geneticist, but even I know that those stats are meaningless. They have performed thousands of tests - arguably every nucleotide they sequenced is a test, or at least every variant from reference they examined. If you do a thousand tests then your p value needs to be below 0.05/1000 before it is statistically significant. So this distinction needs to be discussed and made crystal clear, and those chi-squared tests need to be removed.

Thank you for this comment. In our study, we are searching for rare variants described as pathogenic, likely pathogenic or VUS with low frequencies (<1%) in the general population, as well as variants that have been described as hypomorphic potentially associated with adult glaucoma. We agree with the reviewer's opinion, and the chi-squared test can be confusing. We thought it was a way to statistically demonstrate that the allele frequencies observed in our study are higher than those observed in the general population. We do not intend to conduct an association study because we would need a larger number of patients. For this reason, we have removed the column with the frequencies of our study in the Table 3 and the chi-squared test from the manuscript and Table 3.

Additionally, as reviewer suggested, we have added the familial segregation of some variants (see figure 2).

Secondly there is much information missing or in need of clarification. The cohort they study is not well described. The section in mat/mets only talks about the 65 patients, not the 34, so reading that I get the impression the 34 are a subset of the 65, but on lines 97 and 296 these are clearly referred to as a separate cohort. 

Thank you for this comment. As mentioned at the end of the comment, these are two different cohorts. In the "Materials and Methods" section, the first three subsections address the patients included in the study. In the first subsection ("Patients"), we refer to all patients included in both cohorts. In the second subsection ("Study of glaucoma-related genes using NGS"), we only refer to the patients who underwent NGS study. In the third subsection ("Sanger sequencing of CYP1B1 in a new cohort of POAG patients"), we are discussing the patients in whom only the CYP1B1 gene was studied by Sanger sequencing. We have rewritten certain parts of the text in these subsections to make it clearer.

Also it seems bizarre that they would weaken a genetic study by making their cohort so diverse - surely there are plenty of POAG patients across 18 Spanish centres without introducing heterogeneity by including NTG and OHT? 

We excluded patients with OHT alone to ensure that only patients with glaucoma were included, making the groups more homogeneous and avoiding a predominance of patients with OHT in the Sanger group. This exclusion did not affect the results as no variants were found in patients with OHT alone.

All patients are said to be unrelated - was any attempt made to recruit affected relatives where possible, as this would allow segregation studies? 

All detected variants were classified as VUS, and therefore, it is not recommended to perform familial segregation analysis for VUS variants in clinical practice (Pollard et al., Nat. Rev. Genet. 2019). However, given the potential role of CYP1B1 hypomorphic variants in adult glaucoma as described by others in the literature and the availability of samples from relatives, we performed the segregation study of the p.Y81N variant in the relatives of two index samples. We have added this information to the manuscript and a new figure with the pedigrees (Figure 2). 

In the introduction the authors rightly suggest genetic effects may be enriched in patients with early onset or strong family history - was any attempt made to recruit particularly among those groups? 

In the current study, our objective was the study of patients with primary open-angle glaucoma. This is the most common form of glaucoma in the population and typically manifests after the age of 40. Early-onset glaucoma, such as congenital and juvenile glaucoma, were not included in the study as they do not fall within the scope of the research. This point is not clear in the manuscript, so we have added the following clarification: “Patients with early-onset glaucoma who were diagnosed with PCG or JOAG were excluded”.

Also, if there is no overlap between the 65 patients and the possibly additional 34, the differences between them deserve some note. The second group seem milder, with more OHT and less surgery.

As mentioned above, to ensure the homogeneity between the groups and to avoid over-representation of OHT patients in the cohort of patients studied by Sanger, we excluded patients with OHT alone. This exclusion did not affect the results, as no variants were found in patients with OHT alone.

The process by which candidate genes were selected for screening is also poorly described. The authors simply say they looked at Pubmed and chose these 72 genes. Why these and not others, what criteria were used to select candidates? Also when was the selection made, because Pubmed is a rapidly changing and developing resource that may well give a very different set of candidates if they look again now. There are the obvious ones in their list but there are a few for which I was unaware of a link. I did also wonder why a couple of big hits from GWAS were not included - TCMO1, FOXC1, GAS7 and the CDKs are in this list, but not AFAP1 or ABCA1? I'm not asking for a one by one justification, just some objective criteria.

Thank you very much for the comment, which is entirely appropriate, considering that genes associated with glaucoma are constantly being updated. The gene panel was designed based on a thorough search of the OMIM, Orphanet, and HGMD databases, complemented by an exhaustive search on PubMed. For this reason, we have included the references selected in this search. As the reviewer rightly points out, it's very likely that if we were compiling this list of genes today, we would include more genes. However, we think that the study provides valuable insights into the genes that were under investigation at the time and that these findings can serve as a foundation for future research. Nevertheless, some of the genes we reviewed have been excluded after a careful consideration.

There are also gaps in the technical data. How many nucleotides did exons of the 72 genes amount to, how much flanking DNA was included, were 5' and 3' sequences included in the sequencing, how much intronic, 5' and 3' sequence was analysed, what kind of HiSeq was used, what depth of coverage was obtained?

All the exons of the genes included in the study and intronic regions up to +10 bp from the exon were analyzed, with coverage greater than 20x. UTR regions were not analyzed. In addition, our libraries were sequenced using the HiSeq™ 2000 System. We have included this information in the manuscript.

Finally there was very little said about what was found other than the variants cherry-picked for presentation. I think we can safely presume that the nine variants presented were not the only things they found. There needs to be a brief summary statement of everything identified, then a list of everything that passed certain minimal criteria that need to be clarified - below 1% in the population (clear), in the coding sequence and +/-10bp (not clear), non-synonymous (not clear) then presumably predicted VUS, likely pathogenic or pathogenic (again not clear) - should be attached in a supplementary file. Lastly they need some further explanation of why only these nine were chosen for further examination.

As we have mentioned above we have reported rare variants classified/described as pathogenic, likely pathogenic or VUS with low frequencies in the general population (<1%), as well as variants that have been described as hypomorphic variants associated with glaucoma in other studies. The variants classified as benign o likely benign that are not associated with glaucoma are not described due to they are not associated with the disease. All variants reported were found in coding regions, but as we mentioned in the 'Materials and Methods' section, we have also analysed intronic regions up to 10 bp with respect to the exons. 

Besides, to clarify this point, we have included the ACMG classification in Table 3. The variants included in this table are the ones that passed these filters and are therefore candidates for description in the manuscript as they may have some association with adult glaucoma.

In addition I have a few mostly minor corrections. On line 68 and below "Through linkage analysis, 23 loci and four genes.... (ref 5)"; this is clumsy, they seem to be saying at a certain time there were only these findings then later there were more, but the "later" they quote is reference 7 that was actually published in 2016, a year before reference 5?

Thank you for the note. We have removed reference 5 to avoid any confusion.

Line 130 - they need to decide whether they prefer NTG or LTG and stick with one. As far as I know these are not two different diseases but different names for one.

We prefer the use of "low-tension glaucoma," so we have eliminated the use of "normal tension glaucoma" to avoid any confusions.

Legend to table 3 needs to state that (I assume) all variants were heterozygous.

Correct, all variants were detected in heterozygosity. We have added this information in the tables and in the manuscript.

Were mutations found in NGS rechecked by Sanger? Perhaps not essential with very modern NGS machines but if the data were older it might be reassuring, and also allows segregation if other family members are available.

No, the mutations detected by NGS were not rechecked by Sanger sequencing. Sanger sequencing was not demmed necessary. The NGS equipment used is highly accurate, so there is no need for additional validation by Sanger sequencing. Besides, the results of quality controls and coverage were satisfactory. In our routine work, Sanger sequencing is only required to validate frameshift variants and in some cases where the variant appears in less than 30% of the reads. This was not the case for any of the variants reported in the manuscript.

Figure 1 is described in the text as homology "modelling". To me modelling means something different, this is just comparing evolutionary conservation of these residues, and since it only compares across mammals it isn't very conclusive - are these residues conserved in birds, reptiles, amphibians or fish? If not it would be useful to know.

Thank you for this note. Yes, these residues are conserved in species other than mammals. Regarding residue 28, we can see that it is conserved in a multitude of species, except for some bird and amphibian species. As for residue 81, we have observed its presence in all types of species assessed. Additional examples of birds, reptiles, amphibians and fish have been included in Figure 1. Furthermore, we have updated the figure caption to clarify that it is a comparison of residues in different species.

Figure 2d is too small and fuzzy to derive anything from it.

We have enlarged the image for better visibility.

Line 284 - "Adult glaucoma is a multifactorial genetic disease upon which a series of environmental factors must hatch" - clumsy - I suggest "Adult glaucoma is a multifactorial genetic disease the outcome of which is also influenced by a series of environmental factors" - or similar.

Thank you, we have added this clarification to the text.

Also in the discussion there is much talk of hypomorphic alleles - a clear definition of what the authors mean by that would be useful.

Thank you, we have added a definition of hypomorphic alleles in the discussion.

--

Reviewer #2: The authors analysed a cohort of 65 patients with primary open-angle glaucoma (POAG) using a custom gene panel of 72 genes associated with the disease, to test the use of NGS as a diagnostics method. They reported 9 variants in 7 genes in 15% of patients. However some of the genes included in the panel have been reviewed and showed to have no association to POAG and none of the reported variant presented evidence to support a pathogenic impact. Additionally, the cohort is quite small to identify variants that contribute to <10% of the disease, especially with no variants reported in the main contributor to POAG (e.g. MYOC). Therefore, the evidence presented in this study does not support NGS for genetic diagnosis of glaucoma and does not support the conclusions.

Although variants in Mendelian genes account for around 5% of POAG cases, this is mainly driven by MYOC. It is surprising that no variants were reported in MYOC in this study. Have the authors identified any variants in the gene in their cohort?

As the reviewer points out, it is noteworthy that we did not find variants in the MYOC gene in our cohort of adult glaucoma patients. In our centres, we have detected patient with pathogenic variant in MYOC, but these typically involve younger patients, with diagnoses often made in their 20s or 30s. In these cases, the main diagnosis is juvenile open-angle glaucoma. In fact, it is considered that the MYOC gene is primarily associated with juvenile open-angle glaucoma. Patients with congenital or juvenile glaucoma were excluded from our cohorts, and this fact could explain why we did not identify any patient with MYOC variants. However, the MYOC gene was included in the panel.

We have data on younger cohorts of glaucoma patients in which we have found variants in MYOC (data not published yet). However, as mentioned earlier, the age of diagnosis was lower, and the glaucoma signs led to the diagnosis of juvenile open-angle glaucoma, which is not the focus of this article.

Regarding the gene panel, it was designed based on a thorough search of OMIM, Orphanet, and HGMD databases, complemented by an exhaustive search on PubMed. For this reason, we have added the references selected in this search. As the reviewer rightly points out, it's highly likely that if we were to compile this list of genes today, we would include more genes. However, we think that the study provides valuable insights into the genes that were under investigation at the time and that these findings can serve as a foundation for future research. Nevertheless, some of the genes we reviewed have been excluded after a careful examination.

Some of the genes included in the panel such as WDR36 and NTF4 have been reclassified by initiatives conducting evidence-based reviews (e.g. ClinGen & PanelApp) as having no association with POAG. Similarly, some rare variants in genes such as EFEMP1 and TEK have been associated with JOAG/POAG in a monogenic manner but have not been included in the list. The list of genes associated with POAG needs to be revised for proper interpretation of results in the context of using NGS as a diagnostic strategy.

Specifically, for WDR36 the initial variants reported in the literature were found to be too common in the general population, a meta-analysis showed no association in case-controls, the gene is ubiquitously expressed therefore expression in eye tissues does not constitute supporting evidence and no functional evidence inc. animal models reported a glaucoma phenotype. However, none of this is presented in the discussion, which only highlights evidence for an association to POAG. This gene has now been reclassified as having no association and should not be included in a gene panel for POAG assessing NGS for diagnostics.

We agree with the reviewer. EFEMP1 and TEK are genes associated with glaucoma. Regarding EFEMP1, it is a gene with a very recent association, which OMIM has not yet included. TEK is a better-known gene. However, both are associated with juvenile and congenital glaucoma, respectively, so we would not have expected to find variants in these genes. Nevertheless, they are genes associated with glaucoma and should have been included in the panel.

Regarding NTF4, the first study that linked NTF4 and glaucoma was conducted by Pasutto et al. (Am. J. Hum. Genet. 2009). The authors examined the postmortem retina of a patient with no history of eye disease and used in situ hybridization to confirm a specific NTF4 signal localized to the ganglion cell layer of the retina. However, a subsequent study revealed that NTF4 does not play a significant role in the pathogenesis of POAG (Liu et al., Am. J. Hum. Genet. 2010). Nevertheless, since then, we can find different studies with contradictory results and multiple reported cases of glaucoma patients with NTF4 variants (Jemmeih et al., Ophthalmic Epidemiology, 2022; Kumar et al., Gene, 2016; Vithana et al., Mol. Vis. 2010; Huang et al., Sci. Rep. 2018). Besides, if we search for the NTF4 gene in OMIM or Orphanet, it appears associated with POAG. Thus, a meta-analysis of different studies in glaucoma patients with population-based controls would be required to clarify the role of NTF4 in glaucoma. In our case, until this is resolved, we have decided to include the gene in the gene panel. In our cohort, we did not detect any variant in NTF4.

Regarding the inclusion of WDR36 in our panel, clinical studies have yielded contradictory results. Some of these studies indicate a lack of effect of WDR36 in certain populations (Kramer et al., Arch. Ophthalmol. 2006; Frezzotti et al., Br. J. Ophthalmol. 2011; Liu et al., Medicine. 2017). However, other research has demonstrated that WDR36 is a contributing risk factor for glaucoma progression and severity (Footz et al., Hum. Mol. Genet. 2009; Blanco-Marchite et al., Investig. Ophthalmol. Vis. Sci. 2011; Huang et al., Investig. Ophthalmol. Vis. Sci. 2014; Mookherjee et al., Mol. Vis. 2011; Meer et al., Genes 2021). In fact, there is a known genetic interaction between WDR36 and glaucoma susceptibility. It also plays a significant role in retinal homeostasis. Therefore, given the potential of the WDR36 gene as a causative gene for glaucoma, its investigation in these cases is important, although further studies are needed to fully understand its role in glaucoma. To clarify this point, we have added the relevant explanation about WDR36 in the discussion and had to include some additional references.

Most of the genes included in the panel have been associated with POAG or its endophenotypes in genome-wide association studies, however rare variants have not yet been associated with the disease for most genes. The rare variants identified in these genes need to be discussed in this context, highlighting that additional studies, including functional evidence, would be needed before these addition of these genes to gene panels is justified.

Indeed, the reviewer is correct. We have further discussed the variables, and, as expected, the variants are classified as VUS. Most of them have not yet been associated with glaucoma, thus emphasizing the need for additional studies to confirm these findings.

A definition of hypomorphic variants is missing. In general, hypomorphic alleles refer to variants that can be common, do not cause disease on their own but functional evidence supports an impact. It is unclear from the paper which of the variants identified are rare vs hypomorphic, with evidence to support classification.

Thank you, we have added a definition of hypomorphic alleles in the discussion. Currently, as we mentioned in the discussion, there are only hypomorphic variants described in CYP1B1 and SIX6, and we have detected some of them in this study. Regarding rare variants, these will be those classified as pathogenic, likely pathogenic, or VUS and that are infrequent in the population, whose effects are not fully understood. We have also added this information to the manuscript.

Please specify what in silico tools were used for variant impact prediction.

The in silico tools used were: CADD, Polyphen2, DEOGEN2, MutPred, FATHMM-XF, Mutation assessor. MVP, PROVEAN, EIGEN, LRT, SIFT, BLOSUM, DANN, LIST-S2, M-CAP, MutationTaster and PrimateAI. We have added this information to the manuscript.

In addition to the above comments, the limitations need to discuss the detection (or lack of) of CNVs using NGS and its impact on detecting deletions or duplications associated with glaucoma (e.g. TBK1 duplications are known to cause POAG).

We thank you for this comment, and we have included this limitation in the study, and we have discussed it in the Discussion section.

Other comments:

Some references are missing in the text while others reference incorrect papers. line 55 reference 1 does not report on the rate of undiagnosed glaucoma. line 69 reference 5 does not report on the loci and genes associated with POAG. line 86 ref 10 reports on optic disc parameters but no references are provided for the endophenotypes listed (AL, ACD, CCT). line 303 ref 27 and 28 did not report the first association of CYP1B1 to PCG. 

The reference 1 was incorrectly placed in the text. We have now placed it in the correct location. With regard to reference 5, we have removed this reference.

Regarding the line 86, we have added the following reference: Asefa NG, Neustaeter A, Jansonius NM, Snieder H. Heritability of glaucoma and glaucoma-related endophenotypes: Systematic review and meta-analysis. Surv Ophthalmol. 2019;64(6):835-851.

Regarding the sentence with references 27 and 28, we did not intend to imply that these references report the first association of the CYP1B1 gene with PCG. Our intention was to emphasize that the CYP1B1 gene has always been associated with the genetics of primary congenital glaucoma, not with POAG. We have rewritten the sentence to clarify this point.

There are no references provided for the prevalence of glaucoma (line 53), the familial risk (line 75), the gene contribution to POAG (line 78), the 100 loci reported for POAG (line 84).

The reference for the sentence in line 53 is reference 1, which was incorrectly placed. We have now placed it in the correct location.

Regarding the other lines, as the reviewer recommends, we have added the following references:

• Line 75: 

o Le et al. Risk factors associated with the incidence of open-angle glaucoma: the visual impairment project. Invest Ophthalmol Vis Sci. 2003:44(9):3783-9.

o Green et al. How significant is a family history of glaucoma? Experience from the Glaucoma Inheritance Study in Tasmania. Clin Exp Ophthalmol. 2007;35(9):793-9.

• Line 78: Rao et al. Complex genetic mechanisms in glaucoma: an overview. Indian J Ophthalmol. 2011;59(Suppl. 1):S31-42.

• Line 84: Han et al. Large-scale multitrait genome-wide association analyses identify hundreds of glaucoma risk loci. Nat Genet. 2023;55(7):1116–25.

Table 3:

- Please note that the use of the ACMG PP5 and BP6 criteria has been discontinued (PMID 29543229) and should be removed for CYP1B1 Y81N.

Thank you for the notice. We have removed both criteria from the variant. Additionally, we have reviewed the classification of this variant, and the correct criteria are: PS3, PP3, BS1, and BS2. We have added the PS3 criterion because there is a study (Lopez-Garrido et al., Clin Genet. 2010) that conducts a functional analysis of the Y81N variant. The authors reported that the presence of six variants in the CYP1B1 gene (including the Y81N variant) reduces enzymatic activity, with activity ranging from 18% to 40% of the wild-type protein. Therefore, the variant is classified as a VUS, which has also been described as a hypomorphic variant.

- PVS1 only applies when LoF is a known disease mechanism. Therefore it does not apply to OPTN variants which cause POAG through a gain of function mechanism since LoF variants such as nonsense are not expected to cause glaucoma. Therefore, there is no evidence to support a role of the nonsense OPTN variant reported with POAG. ClinGen recently reviewed the association of OPTN to POAG and concluded that the only variant showing an association is E50K (https://search.clinicalgenome.org/kb/gene-validity/CGGV:assertion_1d8edc80-413c-4c76-bf06-89198175ac65-2022-05-10T170000.000Z)

Thank you for the comment, we agree with what the reviewer has stated. It is indeed a loss of function and should not be classified under the PVS1 criterion. We have removed it from the table, and the variant is still classified as a VUS. Loss of function in the OPTN gene is associated with amyotrophic lateral sclerosis (ALS), and variants predominantly described in association with glaucoma are missense mutations. Nevertheless, there are cases where ALS and glaucoma coexist, as seen in one of the patients reported by Mayurama et al. (Nature, 2010). Although recent studies have reported that E50K is the only variant known to be causative of glaucoma associated with the OPTN gene, the presence of variants in this gene may contribute to some added risk for glaucoma in certain patient populations, as indicated by Fox and Fingert (Progress in Retinal and Eye Research, 2023) in their latest review. For all these reasons, the variant found in our study should be considered a VUS.

We have included this information in the text, and we have also added the two references mentioned.

- All in silico tools (CADD, SIFT, PP2, REVEL, Mutation Taster) predict a benign impact for CARD10 T436M which should then meet BP4. Please revise and acknowledge in the discussion.

We have added the BP4 criterion in the table. Nevertheless, the variant continues to be classified as a VUS. We have added this information to the manuscript.

- Please report the highest AF in a population instead of the general population as some of these variants are more common in some populations than others, which should be used when classifying variants.

- Add the ACMG classification for each variant based on evaluation in Table 3.

We have added the highest AF and the ACMG classification for each variant to the Table 3.

- Move nucleotide nomenclature to Table 3 instead of Table 4.

We have made the change to move nucleotide nomenclature to Table 3 instead of Table 4.

- CYP1B1 S28W is missing from Table 3.

Originally, Table 3 only displayed the results of the NGS study. The p.S28W variant in CYP1B1 was found in the Sanger study. To clarify this point, we have decided to include all variants in Table 3.

Table 4:

- OPTN R298H in Patient 8 should be Q518*.

It is absolutely right; it was a mistake in the table preparation. We have already corrected it.

The prevalence of glaucoma in 2040 reported by Tham et al. is 112 million, not 120 million.

That's correct. The prevalence of glaucoma in 2040 reported by Tham et al. is 112 million, not 120 million. We have already made the change in the manuscript.

line 127 the range of age at diagnosis is listed as 25-76y but line 129 reports a patient diagnosed at age 18y.

Thank you for this observation. The age range for patients with POAG is indeed 18 to 76 years. It has now been corrected.

line 267 CYP1B1 S28W is in gnomAD, please correct.

We have already corrected this data. The variant has a very low population frequency (0.004%).

--

Reviewer #3: The authors have screened two different cohorts of POAG patients for disease causative mutations by using two different techniques; NGS panel for 72 genes associated with POAG and Sanger sequencing for Cyp1b1, a frequently mutated gene in patients with glaucoma of different types. The main aim is the applicability of custom NGS panel for POAG. The study is well planned and methods are up to date. The results are as per expectations but the interpretations need some more work to reach a definite conclusion. 

Following are some points for improvement of the draft

1. For first cohort of 65 patients, subjected to NGS, 53 patients have family history. It is important to check segregation of the alleles in other family members, to confirm its association with the disease.

2. The most common gene was CYP1B1 as per results. The non-penetrance of various CYP1B1 alleles is reported and its diagnosis in a proband need segregation studies as well to establish its pathogenic role.

All detected variants were classified as VUS, and therefore, conducting studies on relatives of VUS variants whose association is not fully understood is not recommended in clinical practice. However, due to the role of CYP1B1 in glaucoma and the description of hypomorphic variants in the literature, we decided to perform the segregation study of the Y81N variant in the relatives of the patients detected in the NGS study. We have added this information to the manuscript and a new figure with the pedigrees. Relatives of the patients detected in the Sanger study were not available for the segregation study.

3. Why CARD10 gene was not selected for sanger sequencing as it is second common gene as per NGS results

The CARD10 gene was not selected for Sanger sequencing because, despite being the second most common gene according to NGS results, we decided to study CYP1B1 due to its well-documented nature. Its association with glaucoma is more established, and in scientific literature, we can find hypomorphic alleles described in relation to POAG. Furthermore, the cost of studying CARD10 by Sanger sequencing is 10 times more expensive than the cost of studying CYP1B1 (CARD10 has 20 exons, whereas CYP1B1 has only 2 exons). However, we are open to revisiting this decision based on the significance of the CARD10 gene in further analyses.

4. Authors should discuss the findings in the light of already reported data of the glaucoma genes in Spanish patients

To our knowledge, there are previous studies with Spanish populations affected by pseudoexfoliative glaucoma, primary congenital glaucoma, juvenile glaucoma and anterior segment dysgenesis. Regarding POAG, there is a study from 2007 that retrospectively investigated the contribution of myocilin (MYOC) and optineurin (OPTN) to a cohort of OHT and POAG Spanish patients. They found that a minority of adult-onset high-pressure POAG patients carried heterozygous disease-causing mutations in the MYOC gene and that OPTN was not involved in either OHT or POAG. (Lopez-Martinez F et al. Mol Vis. 2007 Jun 14;13:862-72.). The same authors published another article several years later and found heterozygous hypomorphic CYP1B1 mutations in Spanish POAG patients (López-Garrido MP et al. Clin Genet. 2010 Jan;77(1):70-8). 

--

Reviewer #4: I would like to make several recommendations which the authors may find useful to improve their paper:

• As the authors pointed out several times the importance of genetic diagnosis of glaucoma, I would recommend classifying detected variants (e.g. in table 3, not only ClinVar data). Additionally, in the same table, the ACMG criteria are listed in the column “Predicted effect in silico”. It is not clear are those criteria taken from Varsome database or the authors selected those by themselves (which I would recommend).

Thank you for the comment. We have clarified this in the table and added a column with the ACMG classification of the detected variants.

• Lines 77-78. Missing references.

We have added the following reference: Rao et al. Complex genetic mechanisms in glaucoma: an overview. Indian J Ophthalmol. 2011;59(Suppl. 1):S31-42.

• Lines 191-192. What variants were found correlated?

The presence of thin pachymetry was detected in two patients (patient 1 and patient 8). The variant p.Y81N in CYP1B1 was detected in patient 1, while the variant p.R298H in OPTC was detected in patient 8. As discussed, the presence of thin pachymetry readings is a recognized endophenotypic trait that amplifies the severity of glaucoma. Nevertheless, further studies are warranted to explore this relationship more comprehensively.

• The variant c.83C>G (p.Ser28Trp) in the CYP1B1 gene was not presented in tables 3 and 4.

Thank you for this comment. The variant has already been included in the tables.

• Lines 255-260. It is misleading how many variants were detected in CYP1B1 (2 in the tables, here one more, 4 in discussion...)

• Lines 307-309. Four patients with variants in CYP1B1?

The variant p.Y81N was detected in four patients (two in the NGS study and two in the Sanger study). Additionally, in the Sanger study, we found one patient with a different variant in the CYP1B1 gene. Therefore, as shown in Table 4, a total of five patients with variants in CYP1B1 were identified (four patients with p.Y81N and one patient with p.S28W).

---

## [Decision Letter · Decision Letter 1]

22 Nov 2023

Next-generation sequencing-based gene panel tests for the detection of rare variants and hypomorphic alleles associated with primary open-angle glaucoma

PONE-D-23-03532R1

Dear Drs. Milla,

We’re pleased to inform you that your manuscript has been judged scientifically suitable for publication and will be formally accepted for publication once it meets all outstanding technical requirements.

Kind regards,

Alvaro Galli

Academic Editor

PLOS ONE

Additional Editor Comments (optional):

Reviewers' comments:

Reviewer's Responses to Questions

**Comments to the Author**

1. If the authors have adequately addressed your comments raised in a previous round of review and you feel that this manuscript is now acceptable for publication, you may indicate that here to bypass the “Comments to the Author” section, enter your conflict of interest statement in the “Confidential to Editor” section, and submit your "Accept" recommendation.

Reviewer #3: All comments have been addressed

2. Is the manuscript technically sound, and do the data support the conclusions?

Reviewer #3: Yes

3. Has the statistical analysis been performed appropriately and rigorously? 

Reviewer #3: N/A

4. Have the authors made all data underlying the findings in their manuscript fully available?

Reviewer #3: Yes

5. Is the manuscript presented in an intelligible fashion and written in standard English?

Reviewer #3: Yes

6. Review Comments to the Author

Reviewer #3: The authors screened 63 patients/families affected with POAG, the queries raised in 1st review have been responded satisfactorily.

7. PLOS authors have the option to publish the peer review history of their article (what does this mean?). If published, this will include your full peer review and any attached files.

Reviewer #3: No

---

## [Editor Report · Acceptance letter]

8 Jan 2024

PONE-D-23-03532R1 

PLOS ONE

Dear Dr. Milla, 

I'm pleased to inform you that your manuscript has been deemed suitable for publication in PLOS ONE. Congratulations! Your manuscript is now being handed over to our production team.

Kind regards, 

on behalf of

Dr. Alvaro Galli 

Academic Editor

PLOS ONE